# Dose-Response Relationship between Gestational Weight Gain and Neonatal Birthweight in Chinese Women with Excess Weight/Obesity and Gestational Diabetes Mellitus

**DOI:** 10.3390/healthcare11162358

**Published:** 2023-08-21

**Authors:** Jing He, Kaili Hu, Binghua Wang, Zhen Chen, Hui Wang

**Affiliations:** 1Nursing Department, Tongji Hospital of Tongji Medical College of Huazhong University of Science and Technology, 1095 Jiefang Avenue, Wuhan 430030, China; jing26@whu.edu.cn (J.H.); hukaili@tjh.tjmu.edu.cn (K.H.); wangbinghua@tjh.tjmu.edu.cn (B.W.); 2School of Nursing, Tongji Medical College of Huazhong University of Science and Technology, 13 Hangkong Road, Wuhan 430030, China; 3Department of Gynaecology and Obstetrics, Chongqing Health Center for Women and Children (Women and Children’s Hospital of Chongqing Medical University), 120 Longshan Road, Chongqing 401147, China; zhenchen@hospital.cqmu.edu.cn

**Keywords:** gestational diabetes mellitus, overweight, obesity, gestational weight gain, neonatal birthweight, China

## Abstract

Total gestational weight gain (GWG) is identified as a strong and potentially controllable predictor of long-term health outcomes in women with gestational diabetes mellitus (GDM) and infants. When the total GWG of women with excess weight/obesity and GDM does not exceed the Institute of Medicine (IOM) suggested range, neonatal birthweight outcomes may be favorable, but the evidence is limited. Therefore, the objective of this study was to evaluate the dose-response relationship between increased total GWG and the risk of neonatal birthweight in Chinese women with excess weight/obesity and GDM. This study obtained electronic medical records (EMR) from the hospital information system (HIS) of the Chongqing Health Center for Women and Children between July 2017, and June 2020. A retrospective study analyzed the effect of the total GWG of women with excess weight/obesity and GDM on neonatal birthweight. The dose-response relationship between total GWG and neonatal birthweight was studied using a generalized linear model and embedded restricted cubic splines (RCS). The average age of all women with GDM was 31.99 ± 4.47 years, and 27.61% were advanced maternal age (≥35 years). The total GWG among women with excess weight and obesity and GDM greater than the IOM recommendations were found in 42.96% and 58.62% of cases, respectively. Total GWG in women with excess weight and excessing the IOM recommended range is a risk factor for large gestational age (LGA) [adjusted odds ratio (aOR) 0.1.47, 1.08–2.01] and macrosomia (aOR 1.55, 1.04–2.31). In the obesity above group, excessive weight gain increased the risk of LGA (aOR 2.92, 1.33–6.41) and macrosomia (aOR 2.83, 1.03–7.72). We used an RCS to examine pregnant women with excess weight and GDM and discovered a linear dose-response relationship between total GWG and LGA/macrosomia. In women with excess weight and obesity, increases in total GWG above the lowest end of the IOM recommendations range (7 kg and 5 kg) were associated with an increased risk of LGA and macrosomia. Therefore, research is urgently needed to support maternal and newborn health to provide recommendations for the ideal weight increase in women with excess weight/obesity and GDM.

## 1. Introduction

Weight gain and obesity are serious global health problems and independent primary risk factors for gestational diabetes mellitus (GDM) [1]. Women with pre-pregnancy excess weight or obesity may experience increased insulin resistance during pregnancy and exacerbate hyperglycemia, which can be further aggravated by excessive weight gain [2,3]. GDM is primarily diagnosed in the second and third trimesters of pregnancy and is defined as any level of glucose intolerance initially detected during pregnancy [4]. Traditional approaches to GDM management have centered on monitoring and treating it throughout pregnancy to control hyperglycemia in pregnant women. Studies suggest adequate glycemic control in pregnant women can prevent poor outcomes for mothers and neonates [5,6,7]. However, some studies have found that women with excess weight/obesity and blood glucose levels in the normal range have higher large gestational age (LGA) outcomes than women with average or low pre-pregnancy weight [8,9]. Furthermore, excess weight/obesity and GDM may exacerbate adverse neonatal effects, especially the LGA and macrosomia [10,11].

Many factors influence GDM and unfavorable pregnancy outcomes, including pre-pregnancy excess weight/obesity, proper glycemic control, treatment modalities (diet and physical exercise alone or in combination with medication), changes in the gut microbiota, and gestational weight increase [3,12]. Komen et al. found that total weight change was associated with poor blood glucose control, cesarean section, and excessive amniotic fluid [13]. According to Hedderson et al., substantial weight gain may increase the risk of poor glycemic control during pregnancy and postpartum in women with GDM [14]. The Institute of Medicine (IOM) suggested updated guidelines for GWG (gestational weight gain) in 2009. Still, they did not address GWG in high-risk pregnancies or whether more stringent requirements for weight gain could reduce the adverse effects of hyperglycemia [15]. The synergistic effect of excessive weight gain and blood glucose control is essential for enhancing adverse pregnancy outcomes with GDM.

Goldstein et al. discovered that GWG was significantly higher than the IOM recommendations for body weight increase in 47% of pregnancies in a meta-analysis involving approximately 130,000 pregnant women studying GWG and pregnancy outcomes [11]. The IOM has published guidance for appropriate weight gain during pregnancy for healthy pregnant women in each preconception body mass index (BMI) category [15]. And the recommendations are based on the weight gained during pregnancy to achieve the ideal neonatal birthweight [15]. Studies discovered, however, that women with GDM and pre-pregnancy obesity were more likely to exceed the IOM’s suggested criteria [16,17]. Furthermore, studies have revealed that the total GWG of women with GDM is below the IOM recommendations value can reduce adverse pregnancy outcomes, and are favorable to postpartum metabolism [18,19].

GWG has been identified as a strong and potentially controllable predictor of short-and long-term health outcomes in women and infants with GDM, increasing the risk of LGA and macrosomia [18,19]. Mustaniemi et al. reported that GDM, obesity, and excess GWG are associated with a higher risk of LGA in infants, and interventions targeting normal GWG are also likely to reduce LGA rates and birthweight standard deviation scores [20]. However, another study in the Chinese population found that GWG was advantageous for neonatal birthweight outcomes in obese women with GDM below the range suggested of IOM criteria [21]. Additionally, there is a pathophysiological mechanism by which insulin resistance is also impacted by obesity and high total GWG. More glucose crosses the placenta and reaches the fetus due to increased maternal insulin resistance, leading to macrosomia or LGA [21].

In recent years, there has been an upsurge in fetal overgrowth, which may offer a significant risk for future juvenile obesity and diabetes. Fetuses of women with GDM are more likely to have LGA/macrosomia [22]. The current study provides little data on ideal gestational total weight increase in overweight/obese women with GDM and its impact on neonatal birthweight. There is an urgent need to give women with excess weight/obesity and GDM proper weight counseling to decrease total GWG and keep neonatal weight within the normal range. Furthermore, there is a lack of research on the association between sequential changes in weight increase in overweight/obese and GDM women and neonatal weight. To ascertain the impact of total GWG on neonatal outcomes for women with excess weight/obesity and GDM, both below and above IOM recommendations, was the primary objective of this study. Further research was done on the dose-response association between total GWG and the probability of having a low newborn weight.

## 2. Materials and Methods

The Chinese Clinical Trial Registry (ChiCTR) registered this study under registration number ChiCTR2000040588 on 3 December 2020. Furthermore, this study was conducted following the statement of the Report on Enhanced Observational Studies in Epidemiology (STROBE) [23].

### 2.1. Procedures and Participants

The retrospective study was conducted by Obstetric Department of the Chongqing Health Center for Women and Children Between 1 July 2017, and 30 June 2020. And we acquired clinical data (electronic medical records) about women with excess weight/obesity, GDM, and neonates. This study included pregnant women older than 18 years old, had pre-pregnancy BMI that were overweight or obese, were diagnosed with GDM during pregnancy, and reported single and live birth outcomes for the mother and the newborn. The present study excluded pregnant women with type 1 and type 2 diabetes mellitus (T2DM) diagnosed before pregnancy, the absence of core information on total gestational weight gain (GWG), and hereditary diseases of neonates. Furthermore, the presence of infectious diseases in the mother was excluded. This study reviewed three years of clinical data from a single center.

### 2.2. Data Collection Procedures

We obtained clinical parameters and laboratory test results from the hospital information system (HIS) of pregnant women diagnosed with GDM and their paired newborns. We entered them into a standardized template database. We collected data from medical records using the double-entry method to ensure data accuracy. All patient information is anonymized and numbered instead, with no direct patient participation. The data collected included: maternal information (maternal age, pre-pregnancy BMI, gestational number, gestational number and family history of T2DM), maternal, perinatal outcomes (results of three OGTT tests, insulin use, preeclampsia, and delivery mode), and neonatal-perinatal outcomes (gestational age, neonatal gender, weight, hypoglycemia, and hyperbilirubinemia).

### 2.3. Diagnosis and Definitions

Gestational diabetes mellitus was diagnosed based on Chinese guidelines for diagnosis and treatment of GDM with a slightly higher fasting threshold (75 g oral glucose tolerance test (OGTT): fasting plasma glucose (FPG) ≥ 5.1 mmol/L, 1-h glucose ≥ 10.0 mmol/L, 2-h ≥ 8.5 mmol/L) [24]. GDM is often diagnosed between weeks 24 and 28 but can occur at any point during pregnancy [4]. Obstetricians, dietitians, and diabetes educators provide initial (group or individual) education for women with GDM. Then, diabetes educators and endocrinologists reviewed women’s blood glucose records every one to two weeks. The goals of blood glucose management were the targets (FPG < 5.3 mmol/L and 2-h postprandial glucose < 6.7 mmol/L) [24]. Insulin therapy is required when dietary and exercise control glucose instability is measured at a given time of day for seven days, and the glucose level is three or more times above the target value. Metformin was not prescribed.

Height was measured during the first antenatal care. Women were asked to recall their pre-pregnancy weight. On admission, weight at delivery was measured. BMI is equal to weight/height^2^ (kg/m^2^). According to the Chinese adult BMI definition standard, a BMI of 24–27.9 kg/m^2^ is considered excess weight, and a BMI of ≥28 kg/m^2^ is considered obesity [25]. Total GWG is the weight at delivery minus the pre-pregnancy weight. The pre-pregnancy BMI of Chinese women and the IOM-recommended ranges were used to develop guidelines for total GWG. The IOM guidelines recommend a total gestational weight gain of 7.0–11.5 kg for women with excess weight and 5.0–9.0 kg for women with obesity [15]. Based on birthweight, neonates were classified as large for gestational age (LGA) (90th percentile) or small for gestational age (SGA) (10th percentile) using the birthweight curve of Chinese neonates for gestational age [26]. The neonatal hypoglycemia (NH) standard was 2.6 mmol/L [27]. Macrosomia was defined as a birthweight greater than or equal to 4000 g.

### 2.4. Statistical Analysis

Women with GDM and excess weight/obesity were divided into three groups based on total GWG and IOM recommendations: below, within, and above the IOM. Frequency (percentage) for categorical data and mean ± standard deviation (SD) or median (IQR) for continuous data were used to create descriptive information. The independent sample student *t*-test was used for constant data consistent with normal distribution, while the Mann-Whitney U-test was used for inconsistent with normal distribution. And the Chi-squared test was used for categorical data. Without any addressing, all variables have less than 10% missing data. The findings of all statistical tests were statistically significant at *p* < 0.05. The low prevalence of SGA and LBW were 4.11% (80/1945) and 1.03% (20/1945), respectively, which were not included in further analysis.

The relationship between total GWG (reference, within) and neonatal birthweight was evaluated using multivariate logistic regression. This study used regression analysis to examine the potential impacts and variations of various BMIs on the outcomes of LGA and macrosomia in pregnant women with GDM and excess weight/obesity. Results were expressed as odds ratio (OR) and 95% confidence interval (CI), with *p* < 0.05 considered statistically significant. This study identified potential confounders based on univariate analysis, directed acyclic graphs (DAG), and research background, including maternal age (years), gestational age (weeks), gravidity (times), parity (times), family history of diabetes mellitus (yes/no), 75-g OGTT FPG (mmol/L), 1-h glucose (mmol/L), 2-h glucose (mmol/L), cesarean section (yes/no), macrosomia (yes/no), and neonatal gender (male/female) [3,28]. Model 1 has no covariate adjustment; Model 2 was adjusted for the confounding factors identified.

Logistic regression outcomes did not show a linear trend between the total GWG of women with GDM and excess weight/obesity and the risk of LGA and macrosomia outcomes. At the same time, Hedderson et al. suggest that women with GDM and excess weight/obesity are at higher risk for unfavorable neonatal outcomes, and their total GWG should probably not exceed the lowest value of the IOM suggested range [14]. The restricted cubic spline model can show the effect of small continuous changes in weight gain on the odds ratio value of the LGA/macrosomia outcome in the form of a continuous curve [29]. Therefore, we used a generalized linear model incorporated with a restricted cubic spline curve (RCS), the relationship between total GWG and risk of LGA and macrosomia was evaluated [30]. The nonlinear relation is considered in the nonlinear test with *p* < 0.05, and the nonlinear problem is analyzed using the RCS. Then a four-piecewise generalized linear model is established to calculate the inflection point. Knots were defined based on the 10th, 50th, and 90th percentiles of the total GWG distribution. Three knots of total GWG were used in the smooth curves, with excess weight (5, 11, 17) and obesity (4, 10, 18). The reference point was 7 kg for women with excess weight and 5 kg for women with obesity. The same covariates were included in this analysis, whereas all statistical analyses were completed using Stata (v.17.0; College Station, TX, USA).

### 2.5. Ethical Consideration

This study was approved (Approval No. 2020-022) by the Ethical Committee of Chongqing Health Center for Women and Children, which waived the need for informed consent.

## 3. Results

### 3.1. Participant Characteristics

A total of 1945 eligible participants were included in the study. Women were included with GDM and pre-pregnancy excess weight (n = 1597) or obesity (n = 348) (Figure 1). The average age of all women with GDM was 31.99 ± 4.47 years, and 27.61% were advanced maternal age (≥35 years). 52.39% of the pregnant had primiparas. 42.96% (686/1597) of women with excess weight and 58.62% (204/348) of women with obesity exceeded IOM recommendations. Maternal demographic statistics and obstetric characteristics are shown in Table 1, including their age, pre-pregnancy BMI, gestational weeks, gravidae, parity, and mode of delivery.

### 3.2. The Impact of Total GWG on Neonatal Birthweight

In women with excess weight/obesity and GDM, the incidences of LGA and macrosomia were 15.22%/20.69% and 8.70%/10.92%. The findings of univariate analysis of women with GDM, excess weight, and obesity are presented in Table 2 and Table 3, respectively. Since the frequency of SGA was low, and the univariate analysis was not statistically significant, additional inquiry was not carried out. Table 4 demonstrates the findings of a regression analysis of the effect of total GWG on the risk of LGA and macrosomia in women with excess weight and obesity, as well as GDM.

Total GWG in women with excess weight and below the IOM recommended-range is a protective factor for both LGA [adjusted odds ratio (aOR) 0.47, 0.29–0.76] and macrosomia (aOR 0.44, 0.23–0.84). Furthermore, excessive weight gain was a risk factor for LGA (aOR 0.1.47, 1.08–2.01) and macrosomia (aOR 1.55, 1.04–2.31) in the excess weight above group. No significant association was observed between low weight gain and the risk of LGA and macrosomia in women with obesity and those below IOM guidelines. However, excessive weight gain increased the risk of LGA (aOR 2.92, 1.33–6.41) and macrosomia (aOR 2.83, 1.03–7.72) in the group with obesity and total GWG above IOM guidelines.

### 3.3. Dose-Response Relationship between Total GWG and Risk of Neonatal Birthweight

Total GWG and risk of LGA. The study demonstrated that an increase in total GWG was associated with an increased risk of LGA in overweight and obese women (Figure 2a,c). Additionally, women who were overweight and had GDM exhibited a risk of LGA that was 1.63 (1.22–2.11) higher when their total GWG was 11.5 kg (Figure 2a). And, when the total GWG of women with excess weight was less than 7 kg, the lower the weight gain, the greater the protective effect on LGA. We found an overall J-shaped curve for total GWG, and the LGA in pregnant women with obesity, but only between 15.0 (aOR 1.95, 1.01–3.76) and 21.0 kg (aOR 2,37, 1.00–5.61) was statistically significant (Figure 2c).

Total GWG and risk of macrosomia. The findings demonstrated that the risk of macrosomia was positively correlated with the total GWG among pregnant women with GDM and excess weight. However, no discernible impacts were discovered for weight increases of less than 7 kg (Figure 2b). In women with obesity, the total GWG and macrosomia were inverted J-shaped curves. Weight gain of less than 5 kg is protective against macrosomia. On the contrary, the risk of macrosomia increased continuously with a weight gain of 5–13 kg, and the aOR value was between 4.38 and 5.12 after a weight gain of 13 kg (Figure 2d).

## 4. Discussion

This retrospective study of hospital clinical data found that the association with neonatal birthweight was estimated based on BMI characteristics of pregnant women with GDM and excess weight/obesity in China and weight gain recommendations given by IOM guidelines. The findings of this study showed that total GWG below IOM recommendations lowered the risk of LGA and macrosomia and remained steady following the correction model, regardless of pre-pregnancy excess weight or obesity. The neonatal birthweight (LGA/macrosomia) was higher in pregnant women with GDM and obesity who exceeded IOM guidelines. The total GWG range in women with GDM and excess weight or obesity has been studied recently, although no adequate guidelines are still available [31].

Ke et al. reported that excessive weight gain in excess weight/obese women is associated with a higher risk of LGA and macrosomia [32]. In a study by Ping et al., a retrospective cohort analysis of GWG for GDM showed that women with GWG above IOM guidelines had a significant risk of cesarean section, macrosomia, and LGA [33]. In addition, meta-analysis by Vats et al. showed that preconception excess weight/obesity was also significantly associated with the risk of macrosomia and LGA [34]. However, our findings contradict current racial and national reports of women with GDM exceeding the IOM guideline range for LGA and macrosomia at high risk, particularly in women who are overweight or obese before pregnancy [35]. The combined effects of abnormal glucose metabolism in women with GWG and GDM on poor birthweight outcomes require more specific and appropriate GWG recommendations based on their metabolic status.

Furthermore, in pre-pregnancy excess weight/obese women, combining GDM and total GWG may increase maternal circulation-free fatty acid and triglyceride levels via pathophysiological pathways [36]. At the same time, excessive weight gain and fat accumulation induce GDM-sensitive biological responses, including insulin resistance, lipotoxicity, and dyslipidemia [37]. These pathophysiological elements may cause the hyperglycemic maternal uterus to accelerate fetal growth excessively. The results highlight the significance of a healthy weight gain for fetal growth and development in pre-pregnancy obese or overweight women with GDM.

In line with what was previously stated by Catov et al., GDM women in this study who were overweight or obese before becoming pregnant were more likely to have a total GWG higher than the IOM-recommended range [38]. The restricted cubic spline model can use a continuous curve to show the effect of changes in weight gain on the LGA/macrosomia outcome odds ratio value. Thus, we further analysis of the dose-response relationship between total GWG and LGA in excess weight/obese and GDM women showed that the risk of LGA and macrosomia arises when the total weight gain in excess weight/obese and GDM women exceeds the lower limit (7 kg/5 kg) of the IOM-recommended range, respectively. The risk of LGA and macrosomia increased along with weight gain. Our findings support a systematic review by Goldstein et al. that found a continuous rather than a threshold connection between excess GWG and newborn weight outcomes [11].

As a result, there was a linear trend in the dose-response data for pre-pregnancy excess weight/obesity, total weight growth of GDM, and neonatal weight, and it may be easier to understand the effects of varied weight gain on LGA and macrosomia. Nevertheless, our findings suggest that the range recommended by the IOM may not apply to women with GDM and pre-pregnant excess weight/obesity. Xu et al. studied using modified GWG targets (2 kg less from the upper and lower limits of the IOM recommended range) for lower LGA and macrosomia incidence than the original IOM targets [39].

Furthermore, women with obesity and GDM who have total GWG below IOM guidelines had a reduced risk of macrosomia, which means that the neonatal weight was more indicative of fetal growth [40]. The dose-response results of total weight gain and neonatal weight showed a linear trend, which could more intuitively see the risk of different weight gain on LGA and macrosomia. The outcomes of this study indicate the significance of weight management for pregnant women with GDM who are overweight or obese to reduce weight gain and maintain a healthy weight during pregnancy. Ferreira et al. studied total weight gain below IOM recommendations showing benefits in improving adverse pregnancy outcomes [41]. Therefore, in clinical practice, GDM and weight gain in overweight/obese women should be strictly controlled, and uniform standards for total GWG reduction (below the IOM recommendations) should be formulated to ensure adequate fetal growth.

In contrast, there was no statistically significant difference between weight gains based on IOM and SGA in our study among women who were overweight or obese. In addition, no adverse effects of decreased weight gain on low birthweight outcomes were discovered in research participants who were overweight or obese and had GDM, demonstrating the safety of reducing weight gain [42]. More recently, Huang et al. also showed that women with GDM who had GWG below the guidelines did not increase the risk of SGA, although SGA babies had more adverse outcomes in newborns born to mothers with GDM [43]. Additionally, Drever et al.’s analysis of a clinical picture with decreased preconception BMI, fasting blood glucose levels, and baseline USS growth measurements alone may indicate that women with GDM only need minimal active glucose management to prevent SGA [44]. Furthermore, Monteiro et al. found that only women with normal and low pre-pregnancy BMI and a GWG below 3 kg had a higher risk of SGA [45].

In addition, further proof from pertinent studies is needed to compensate for sample size, population characteristics, and other aspects. In particular, the total GWG data distribution for women with GDM and excess weight/obesity included was small on both sides of the tail sample size in this study. Although, the truncated power basis of the RCS allows linear extrapolation outside of the external nodes. However, extrapolation can still be dangerous when the external nodes are on the tails of both sides of the data. Therefore, the results of this study can provide a basis for the formulation of a reasonable total GWG among women with GDM and excess weight/obesity. But further large-scale, multi-center prospective studies are needed to external validate and recalibrate the appropriate GWG externally.

Besides, total GWG has been associated with a higher risk of poor newborn outcomes in women with pre-pregnancy obesity and GDM [31,46]. However, differences in endocrine metabolism caused by pre-pregnancy excess weight/obesity and heterogeneity in the diagnosis and treatment of GDM may impact all gestational weight gain. At the same time, blood glucose control and pregnancy weight gain are mutually complementary, and the study of Lai et al. showed that more rational weight control is also conducive to the ideal blood glucose control effect (HbA1c < 6%) [47]. Controlling weight gain during pregnancy should be a priority after diagnosis of GDM to optimize pregnancy outcomes and improve postpartum glucose homeostasis, especially in women with excess weight/obese [48]. It is necessary to conduct additional research to investigate phased and revised criteria for weight growth during pregnancy and identify the influence of weight gain during pregnancy on neonatal outcomes in women who have GDM and who are overweight or obese.

This study offers several potential benefits. IOM-recommended criteria were used in the first step of this study, which consisted of performing disaggregated analyses to identify the impact of excess weight/obesity in mothers on the outcomes of newborn birthweight. However, the dose-response relationship between total GWG and adverse neonatal birthweight outcomes was assessed by RCS in this study, which provides a visual understanding of total GWG and the risk of adverse outcomes. Second, this study provides a reference for the formulation of weight gain guidelines according to the characteristics of BMI in Chinese women.

The present study has some limitations. First, the sample size of this study may not be large enough to draw definitive conclusions, and extensive cohort studies of excess weight/obesity who had GDM are needed to support the development of appropriate weight gain guidelines. Second, the sample consisted of Chinese women with excess weight/obese and had gestational diabetes, and ethnic characteristics should be considered and may limit the generalizability of the study findings in other populations. Third, all the information and data used in this investigation came from computerized medical records, and varying measures may yield different results. Fourth, the pre-pregnancy weight of GDM women was estimated using recall and self-reporting, which could introduce bias. Last, this study only analyzed the relationship between total GWG and adverse neonatal birthweight outcomes, and these results need to be viewed critically. The various methods of recording and analyzing GWG data, such as weight gain before the diagnosis of GDM and subsequent weight gain over different gestational periods, can contribute new information to the study’s findings.

## 5. Conclusions

This study found that LGA and macrosomia are dose-responsive to increased gestational weight gain in women who have excess weight or have GDM, and the risk of weight gain exceeding the lower limit of the IOM recommended range was already present. These results suggest that GWG in women with GDM falls short of the IOM-recommended range for beneficial birthweight. Not only is glucose management essential to lowering the risk of macrosomia and LGA, but there is an urgent need for studies to establish suitable weight gain methods for women who are overweight or obese and have GDM. According to the findings, weight management programs should specifically target women with pre-pregnancy excess weight/obesity and GDM to prevent excessive total GWG. Large-scale population studies should be required to evaluate the effects of total GWG reduction in perinatal and postpartum long-term maternal and neonatal outcomes, especially for neonatal birthweight. Meanwhile, it is critical to prevent excess weight and obesity in women of reproductive age to lower the prevalence of GDM associated with obesity and persistent excess weight during pregnancy, improving both maternal and newborn health.

## Figures and Tables

**Figure 1 healthcare-11-02358-f001:**
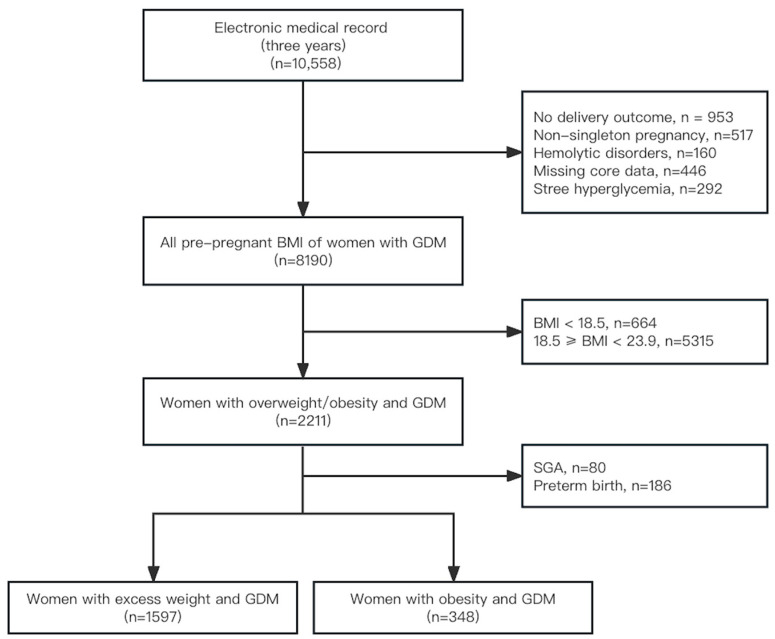
Flow chart for the study population. BMI, body mass index (kg/m^2^); GDM, gestational diabetes mellitus.

**Figure 2 healthcare-11-02358-f002:**
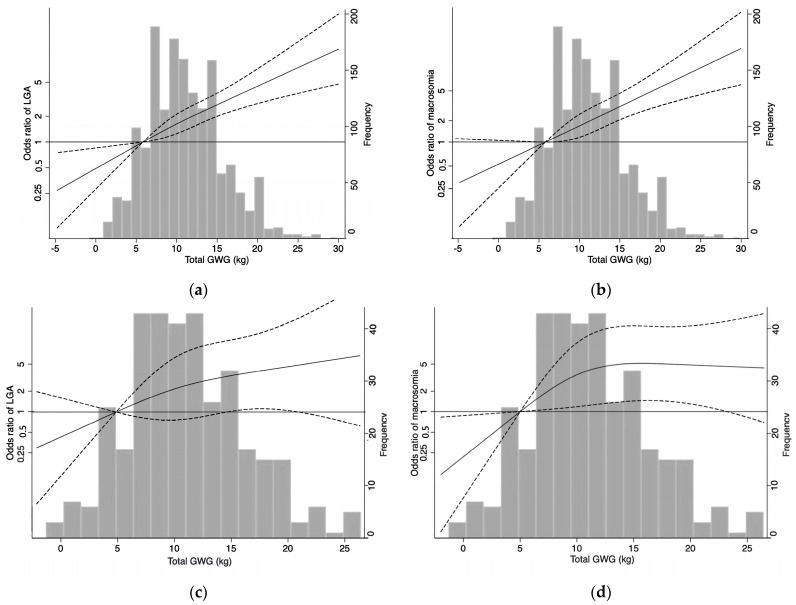
Dose-response relationship between total GWG and risk of LGA and macrosomia. LGA, large for gestational age; GDM, gestational diabetes mellitus; GWG, gestational weight gain; the solid line represents OR; the dotted lines represent the 95% confidence interval (CI); (**a**,**b**) for pregnant women with GDM and excess weight; (**c**,**d**) for pregnant women with GDM and obesity.

**Table 1 healthcare-11-02358-t001:** Characteristics of participants and obstetric information (n = 1945).

Characteristic	Classification	No. (%)
Age, years	18–24	63 (3.24)
	25–29	516 (26.53)
	30–34	829 (42.62)
	35–39	427 (21.95)
	40–44	98 (5.04)
	≥45	12 (0.62)
Pregestational BMI, kg/m^2^	24.0–27.9	1597 (82.11)
	≥28.0	348 (17.89)
Gestational age at delivery, weeks + days	37–37^+6^	207 (10.64)
	38–38^+6^	679 (34.91)
	39–39^+6^	696 (35.78)
	40–40^+6^	351 (18.05)
	41–41^+6^	12 (0.62)
Gravidae	1	546 (28.07)
	2	515 (26.48)
	3	397 (20.41)
	4	250 (12.85)
	≥5	236 (12.13)
	Unknown	1 (0.05)
Parity	0	1019 (52.39)
	1	845 (43.44)
	2	75 (3.86)
	3	4 (0.21)
	≥4	1 (0.05)
	Unknown	1 (0.05)
Delivery mode	Spontaneous delivery	703 (36.14)
	Cesarean section	1242 (63.86)

**Table 2 healthcare-11-02358-t002:** Univariate analysis of women with GDM and excess weight (n = 1597).

Variables	Below (n = 331)	Within (n = 580)	Above (n = 686)	*p*
Age, year	32.33 ± 4.57	32.24 ± 4.37	31.69 ± 4.49	0.051
Gestational age, weeks	38.61 ± 0.91	38.70 ± 0.92	38.62 ± 0.92	0.262
Gravida, times	2.72 ± 1.56	2.61 ± 1.62	2.58 ± 1.54	0.294
Parity, times	0.56 ± 0.59	0.53 ± 0.60	0.49 ± 0.58	0.151
Family history of T2DM	54 (16.31)	82 (14.14)	85 (12.39)	0.229
OGTT fasting, mmol/L	5.05 ± 0.67	5.01 ± 0.81	5.00 ± 0.65	0.298
OGTT 1-h, mmol/L	10.63 ± 1.75	10.37 ± 1.68	10.15 ± 1.80	<0.001 *
OGTT 2-h, mmol/L	9.01 ± 1.67	8.64 ± 1.73	8.50 ± 1.79	<0.001 *
Insulin	40 (12.08)	70 (12.07)	69 (10.06)	0.450
Preeclampsia	11 (3.32)	28 (4.83)	38 (5.54)	0.303
Caesarean section	197 (59.52)	335 (54.76)	474 (69.10)	<0.001 *
Neonatal gender, male	178 (53.78)	294 (50.69)	373 (54.37)	0.399
Macrosomia	12 (3.63)	46 (7.93)	81 (11.81)	<0.001 *
NH	16 (4.83)	34 (5.86)	56 (8.16)	0.087
LGA	24/348 (6.90)	83/602 (13.79)	136/710 (19.15)	<0.001 *
SGA	17/348 (4.89)	22/602 (3.65)	24/710 (3.38)	0.477
NHB	9 (2.72)	20 (3.45)	36 (5.25)	0.102

* *p* < 0.05, Values, mean ± standard deviation by student *t*-test and number (%) by Chi-squared test; T2DM, type 2 diabetes mellitus; OGTT, oral glucose tolerance test; NH, neonatal hypoglycemia; LGA, large for gestational age; LBW, low birth weight; SGA, small for gestational age; NHB, neonatal hyperbilirubinemia.

**Table 3 healthcare-11-02358-t003:** Univariate analysis of women with GDM and obesity (n = 348).

Variables	Below (n = 56)	Within (n = 88)	Above (n = 204)	*p*
Age, year	31.48 ± 4.20	32.34 ± 4.67	31.72 ± 4.51	0.400
Gestational age, weeks	38.44 ± 1.00	38.52 ± 0.93	38.62 ± 0.91	0.705
Gravida, times	2.82 ± 1.62	2.55 ± 1.58	2.75 ± 1.53	0.446
Parity, times	0.51 ± 0.57	0.52 ± 0.55	0.55 ± 0.62	0.926
Family history of T2DM	6 (10.71)	15 (17.05)	34 (16.67)	0.520
OGTT fasting, mmol/L	5.16 ± 0.58	5.24 ± 0.75	5.11 ± 0.58	0.450
OGTT 1-h, mmol/L	10.45 ± 1.61	10.50 ± 1.93	10.51 ± 1.60	0.830
OGTT 2-h, mmol/L	8.52 ± 1.73	8.72 ± 1.88	8.57 ± 1.55	0.813
Insulin	11 (19.64)	13 (14.77)	31 (15.20)	0.688
Preeclampsia	3 (5.36)	8 (9.09)	28 (13.73)	0.163
Caesarean section	32 (57.14)	61 (69.32)	143 (70.10)	0.174
Neonatal gender, male	24 (42.86)	42 (47.73)	96 (47.06)	0.828
Macrosomia	2 (3.57)	7 (7.95)	29 (14.22)	0.045 *
NH	4 (7.14)	10 (11.36)	17 (8.33)	0.621
LGA	9/61 (14.75)	11/92 (11.96)	52/212 (24.53)	0.022 *
SGA	5/61 (8.20)	4/92 (4.35)	8/212 (3.77)	0.197
NHB	1 (1.79)	4 (4.55)	12 (5.88)	0.213

* *p* < 0.05, Values, mean ± standard deviation by student *t*-test and number (%) by Chi-squared test; T2DM, type 2 diabetes mellitus; OGTT, oral glucose tolerance test; NH, neonatal hypoglycemia; LGA, large for gestational age; LBW, low birth weight; SGA, small for gestational age; NHB, neonatal hyperbilirubinemia.

**Table 4 healthcare-11-02358-t004:** Association of total GWG of women with GDM and excess weight/obesity and neonatal birthweight.

Neonatal Birthweight	Pre-BMI and Total GWG	aOR (95% CI)	*p*	OR (95% CI)	*p*
LGA	Excess weight for below IOM	0.465 (0.287–0.755)	0.002 *	0.468 (0.291–0.753)	0.002 *
	Excess weight for above IOM	1.658 (0.812–3.517)	0.331	1.340 (0.517–3.475)	0.547
	Obesity for below IOM	1.470 (1.077–2.006)	0.015 *	1.481 (1.098–1.996)	0.010 *
	Obesity for above IOM	2.920 (1.331–6.406)	0.008 *	2.395 (1.182–4.851)	0.015 *
Macrosomia	Excess weight for below IOM	0.444 (0.230–0.858)	0.016 *	0.437 (0.228–0.837)	0.013 *
	Excess weight for above IOM	1.707 (0.822–3.546)	0.547	0.429 (0.086–2.141)	0.302
	Obesity for below IOM	1.550 (1.043–2.305)	0.030 *	1.554 (1.063–2.273)	0.023 *
	Obesity for above IOM	2.825 (1.034–7.716)	0.043 *	1.918 (0.806–5.561)	0.141

* *p* < 0.05, GWG, gestational weight gain; LGA, large for gestational age; OR, odds ratio, aOR, adjust odds ratio, 95% CI, confidence interval; IOM, Institute of Medicine; pre-BMI, pre-pregnancy body mass index.

## Data Availability

The data presented in this study are available on request from the corresponding author. The data are not publicly available due to ethical, legal, or privacy issues are present.

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
