# Peer review of "Dose-Response Relationship between Gestational Weight Gain and Neonatal Birthweight in Chinese Women with Excess Weight/Obesity and Gestational Diabetes Mellitus"

_healthcare, 2023, doi:10.3390/healthcare11162358_

Round 1

Reviewer 1 Report

The authors present a manuscript which aims to investigate the dose-response relationship between gestational weight gain and neonatal birthweight in Chinese women with excess weight, obesity and gestational diabetes mellitus. The research evaluates the factors related with neonatal birthweight in a developing country where health services for pregnancy and puerperium might have been limited by lack of resources in certain geographical areas. Therefore, the title of the manuscript should be changed as follows: "Dose-response relationship between gestational weight gain and neonatal birthweight in Chinese women with excess weight/obesity and gestational diabetes mellitus" The topic is original and relevant with the field of obstetrics as it addresses a very basic yet usually neglected concept of "the relationship between gestational weight gain and neonatal birthweight" which has crucial significance for undeveloped and developing countries. The similar research in literature mainly focuses on the prevalence of excess gestational weight gain in women with excess body weight, obesity and gestational diabetes while this manuscript aims to specify whether gestational weight gain is directly associated with neonatal birthweight in such women. The methodology of the study is restricted by its retrospective design based on a single official and medical registry. Therefore, the methodology  has been set up as efficiently as possible. Maybe this study should entitled as "retrospective" rather than "cross-sectional" by the authors. The conclusions comply with the evidence presented by the statistical findings and both the conclusions and evidence are all related with the main question and hypothesis of the manuscript. Moreover, all of the references are relevant and up-to-date. All the tables and figures are well drawn and demonstrate the required data. I would recommend to keep all tables and figures. Consequently, I recommend that the manuscript can be accepted for publication in Healthcare after required corrections have been made.    

Author Response

Dear reviewer,

Thank you for the valuable suggestions.

We made the following revisions:

  1. We revised the "Dose-response relationship between gestational weight gain and neonatal birthweight in Chinese women with excess weight/obesity and gestational diabetes mellitus".
  2. We revised and added the “retrospective study” content to the abstract and the methodology.
  3. We revised English language.

Reviewer 2 Report

I found the study very interesting. May be recommendations to politicians and policy makers could be provided as the results are very important. It also could be listed in National Guidelines. I have some little recommendations:

1. The abbreviation as  IOM, LGA  should be explained in first appear in the text  

2. The reference 22: "Furthermore, the study was conducted following the statement of the Report on Enhanced Observational Studies in Epidemiology (STROBE)" is not relevant or the relationship is not visible.

3. Please provide reference: "The diagnosis of GDM usually occurs between gestational 24 and 28 weeks, but it can occur at any time during pregnancy."

4. Please indicate what type of variables: "Samples that met the inclusion criteria and had ten or more variables were included."

5.  I can't found where rate of preterm birth is discussed?

6. May be Table 2 could be reorganized. The presented data is clear but not understandable. It is difficult to found some value to which exactly parameters belongs.

Author Response

Dear Reviewer 2,
Thank you for the valuable suggestions. We have answered and revised every question point-by-point. Please check the PDF for details. Thank you.

Reviewer 3 Report

·         In general, the manuscript requires significant editing by a native English speaker. Please make sure to edit the paper before submitting the revised version.

·         Line 15-34: I didn't understand the entire meaning in abstract. It should be rewritten. Several grammatical errors were found.

·         Line 15: Abstract should start with some brief background information: A sentence or two giving a broad introduction to the study is needed.

·         Line 17-18: “we retrospectively…..neonatal birthweight”- meaning unclear. The method used should be clearly described.

·         Please state the age of participants and date of sampling took place in abstract.

·         Please define abbreviations in the first mention (e.g., IOM, LGA).

·         Please add “China” to the keywords list.

·         Please clearly define GDM in introduction.

·         Line 52-55: What about “gut microbiota”? It is considered an important factor that influenced GDM. I would suggest referring to these articles (Genes (Basel). 2023 Apr 29;14(5):1017; Int J Mol Sci. 2022 Nov; 23(21): 12839; J Diabetes Res. 2021; 2021: 9994734). 

·         Line 75-91: It may be helpful to elaborate on more research gaps that exist with research in GDM and neonatal birthweight. What will this study accomplish? Why this study is important? What is the current situation in China in terms of GDM and neonatal birthweight/diabetes? This should be clarified at the end of introduction. I would suggest that the authors to present the aim of the paper with regards to what is currently known by other studies, therefore highlighting the added value of this study.

·         Line 93-96: Please add more details on how participants were recruited.

·         Line 107-114: Please add more details about data collection.

·         Line 132-133, Line 142: I would suggest clarifying the IOM recommended ranges used.

·         In statistical analysis section, please define both continuous and categorical variables.

·         Table 2: It is unclear the types of statistical analysis used (Chi-squared, student t-test or the Mann-Whitney U-test). This should be clarified.

·         Figure 2 is unclear. I would suggest presenting logistic regression results in table.

·         Line 197-219: Please change text from italic to normal font.

·         The section of the discussion should be qualified with more references.  A discussion should be prepared by organizing information according to: the main findings and comparison of these findings with those reported in the literature; the hypothesis about the non-significantly differences, the strengths and weaknesses of the study and in relation to other studies, information about the present analyses and the implications of this study and future research directions.

·         Please follow the journal guideline for referencing.

The manuscript needs extensive English editing.

Author Response

Dear Reviewer 3,

Thank you for the valuable suggestions. We have answered and revised every question point-by-point. We upload the modified content and reply in a PDF file for your review.

Reviewer 4 Report

I have thoroughly reviewed the manuscript titled "Dose-Response Relationship Between Gestational Weight Gain and Neonatal Birthweight in Chinese Women with Overweight/Obesity and Gestational Diabetes Mellitus". After careful consideration, I regret to inform you that I cannot recommend its publication in its current form due to several significant concerns:

Insufficient Methodological Clarity: The study's methods and statistical analyses are not adequately explained. The use of restricted cubic splines (RCS) for the dose-response relationship warrants a clearer rationale and proper justification.

Lack of External Validation: The manuscript lacks external validation of the findings. To strengthen the study's results, it is essential to validate the dose-response relationship between gestational weight gain and neonatal birthweight in an independent cohort.

Inadequate Sample Size and Generalizability: The sample size in the study might not be sufficient to draw definitive conclusions, particularly given the complexity of the topic. Moreover, the sample consists of Chinese women with overweight/obesity and gestational diabetes mellitus, which may limit the generalizability of the findings to other populations.

Ambiguity in Interpretation: Some of the results are presented ambiguously, making it challenging to interpret the findings accurately.

Recommendations Beyond Study Scope: The manuscript suggests recommendations for developing appropriate weight gain guidelines for women with overweight/obesity and gestational diabetes mellitus. While this is a relevant topic, the study does not provide sufficient evidence or research data to support these recommendations effectively.

Moderate editing is needed 

Author Response

Dear Reviewer 4,

Thank you for the valuable suggestions. We have answered and revised every question point-by-point. We upload the modified content and reply in a PDF file for your review.

Round 2

Reviewer 3 Report

The paper has improved. Few points remain.

Line 259: Please change "statistical analysis" to "ethical consideration".

It seems that the authors delete the presentation of results of Table 2-4. Why? The results should be presented in much more details in the text

Author Response

Dear Reviewer3,

Thank you for the valuable suggestions. We have answered and revised every question point-by-point.

The paper has improved. Few points remain.

Line 259: Please change "statistical analysis" to "ethical consideration".

Response: Thanks for your detailed review. We have revised “statistical analysis” to “ethical consideration” in the Materials and methods Page 4 in line 198.

It seems that the authors delete the presentation of results of Table 2-4. Why? The results should be presented in much more details in the text

Response: Thanks for your detailed review. We have only modified the previous content in terms of language and expression for Table 2-4 of the results, and we adjusted the position of this part of the content.

The specific content is in the Result section on Page 5 in lines 216-232.

The details are as follows:

3.2. The impact of total GWG on neonatal birthweight

In women with excess weight/obesity and GDM, the incidences of LGA and macrosomia were 15.22%/20.69% and 8.70%/10.92%. The findings of univariate analysis of women with GDM, excess weight, and obesity are presented in Tables 2 and 3, respectively. Since the frequency of SGA was low, and the univariate analysis was not statistically significant, additional inquiry was not carried out. Table 4 demonstrates the findings of a regression analysis of the effect of total GWG on the risk of LGA and macrosomia in women with excess weight and obesity, as well as GDM.

Total GWG in women with excess weight and below the IOM recommended range is a protective factor for both LGA [adjusted odds ratio (aOR) 0.47, 0.29-0.76] and macrosomia (aOR 0.44, 0.23-0.84). Furthermore, excessive weight gain was a risk factor for LGA (aOR 0.1.47, 1.08-2.01) and macrosomia (aOR 1.55, 1.04-2.31) in the excess weight above group. No significant association was observed between low weight gain and the risk of LGA and macrosomia in women with obesity and those below IOM guidelines. However, excessive weight gain increased the risk of LGA (aOR 2.92, 1.33-6.41) and macrosomia (aOR 2.83, 1.03-7.72) in the group with obesity and total GWG above IOM guidelines.

Reviewer 4 Report

Thank you for detailed revision of the manuscript titled "Dose-Response Relationship Between Gestational Weight Gain and Neonatal Birthweight in Chinese Women with Overweight/Obesity and Gestational Diabetes Mellitus." The authors have made considerable improvements based on the previous review.

 However, I believe a few minor revisions are necessary to ensure the manuscript's quality and suitability for publication. Specifically, the authors could further clarify the rationale for choosing restricted cubic splines (RCS) for the dose-response relationship analysis and discuss potential strategies to address the lack of external validation data.

 Additionally, it would be helpful to address the implications of the sample size limitation on the study's statistical power and generalizability. Ensuring the results are presented clearly and concisely throughout the results section is crucial.

 Lastly, the authors may want to strengthen the section discussing the implications of their findings for developing appropriate weight gain guidelines and real-world applications of their results.

Minor editing is needed. 

Author Response

Dear Reviewer4,

Thank you for the valuable suggestions. We have answered and revised every question point-by-point. And we have revised and edited the language of the entire manuscript.

Thank you for detailed revision of the manuscript titled "Dose-Response Relationship Between Gestational Weight Gain and Neonatal Birthweight in Chinese Women with Overweight/Obesity and Gestational Diabetes Mellitus." The authors have made considerable improvements based on the previous review.

Response: Thank you for reviewing our manuscript. We have carefully considered your comments and responded to them point-by-point.

However, I believe a few minor revisions are necessary to ensure the manuscript's quality and suitability for publication. Specifically, the authors could further clarify the rationale for choosing restricted cubic splines (RCS) for the dose-response relationship analysis and discuss potential strategies to address the lack of external validation data.

Response: Thanks for your valuable comments.

First, we have added the rationale for selecting restricted cubic splines (RCS) for dose-response relationship analysis in the Statistical analysis section on Page 4 in lines 181-188.

The details are as follows:

“Logistic regression outcomes did not show a linear trend between the total GWG of women with GDM and excess weight/obesity and the risk of LGA and macrosomia outcomes. At the same time, Hedderson et al. suggest that women with GDM and excess weight/obesity are at higher risk for unfavorable neonatal outcomes, and their total GWG should probably not exceed the lowest value of the IOM suggested range [14]. The restricted cubic spline model can show the effect of small continuous changes in weight gain on the odds ratio value of the LGA/ macrosomia outcome in the form of a continuous curve [30].”

Second, we have added in the discussion section that externally validated data may require large cohorts of multi-center prospective cohort studies to externally validate and recalibrate appropriate GWG in the Discussion section on Page 11 in lines 363-372.

The details are as follows:

“In addition, further proof from pertinent studies is needed to compensate for sample size, population characteristics, and other aspects. In particular, the total GWG data distribution for women with GDM and excess weight/obesity included was small on both sides of the tail sample size in this study. Although, the truncated power basis of the RCS allows linear extrapolation outside of the external nodes. However, extrapolation can still be dangerous when the external nodes are on the tails of both sides of the data. Therefore, the results of this study can provide a basis for the formulation of a reasonable total GWG among women with GDM and excess weight/obesity. But further large-scale, multi-center prospective studies are needed to external validate and recalibrate the appropriate GWG externally.”

Additionally, it would be helpful to address the implications of the sample size limitation on the study's statistical power and generalizability. Ensuring the results are presented clearly and concisely throughout the results section is crucial.

Response: Thanks for your valuable comments.

First, we have added to the discussion the statistical power and generalized impact of sample size limitation on the study in the Discussion section on Page 11 in lines 363-368.

The details are as follows:

“In addition, further proof from pertinent studies is needed to compensate for sample size, population characteristics, and other aspects. In particular, the total GWG data distribution for women with GDM and excess weight/obesity included was small on both sides of the tail sample size in this study. Although, the truncated power basis of the RCS allows linear extrapolation outside of the external nodes. However, extrapolation can still be dangerous when the external nodes are on the tails of both sides of the data.”

Second, we have deleted the text “We analyzed and presented the relationship between predicted total GWG and neonatal birthweight using restricted cubic splines (RCS).” in the Result section on Page 5 in line 234, which may be the content of the methodology.

Lastly, the authors may want to strengthen the section discussing the implications of their findings for developing appropriate weight gain guidelines and real-world applications of their results.

Response: Thanks for your valuable comments. We have added and revised the findings for developing appropriate weight gain guidelines and a discussion of practical clinical applications in the Discussion section on Pages 9-10 in lines 333-337 and 344-349.

The details are as follows:

As a result, there was a linear trend in the dose-response data for pre-pregnancy excess weight/obesity, total weight growth of GDM, and neonatal weight, and it may be easier to understand the effects of varied weight gain on LGA and macrosomia. Nevertheless, our findings suggest that the range recommended by the IOM may not apply to women with GDM and pre-pregnant excess weight/obesity. Xu et al. studied using modified GWG targets (2kg less from the upper and lower limits of the IOM recommended range) for lower LGA and macrosomia incidence than the original IOM targets [40].

Furthermore, women with obesity and GDM who have total GWG below IOM guidelines had a reduced risk of macrosomia, which means that the neonatal weight was more indicative of fetal growth [41]. The dose-response results of total weight gain and neonatal weight showed a linear trend, which could more intuitively see the risk of different weight gain on LGA and macrosomia. The outcomes of this study indicate the significance of weight management for pregnant women with GDM who are overweight or obese to reduce weight gain and maintain a healthy weight during pregnancy. Ferreira et al. studied total weight gain below IOM recommendations showing benefits in improving adverse pregnancy outcomes [42]. Therefore, in clinical practice, GDM and weight gain in overweight/obese women should be strictly controlled, and uniform standards for total GWG reduction (below the IOM recommendation) should be formulated to ensure adequate fetal growth.